# Randomized Clinical Trial: Bone Bioactive Liquid Improves Implant Stability and Osseointegration

**DOI:** 10.3390/jfb15100293

**Published:** 2024-10-01

**Authors:** Ashraf Al Madhoun, Khaled Meshal, Neus Carrió, Eduard Ferrés-Amat, Elvira Ferrés-Amat, Miguel Barajas, Ana Leticia Jiménez-Escobar, Areej Said Al-Madhoun, Alaa Saber, Yazan Abou Alsamen, Carles Marti, Maher Atari

**Affiliations:** 1Department of Animal and Imaging Core Facilities, Dasman Diabetes Institute, Dasman 15462, Kuwait; ashraf.madhoun@dasmaninstitute.org; 2Biointelligent Technology Systems SL, C/Diputaccion 316, 3D, 08009 Barcelona, Spain; implant21@yahoo.com (K.M.); eduard.fa@institutferresamat.com (E.F.-A.); miguel.barajas@unavarra.es (M.B.); areejalma222@gmail.com (A.S.A.-M.); asaber@biointelligentsl.com (A.S.); dryazanabualsamen@gmail.com (Y.A.A.); martipages.c@gmail.com (C.M.); 3Periodontology Department, Universitat Internacional de Catalunya (UIC), C/Josep Trueta s/n, 08195 Barcelona, Spain; neuscarriober@gmail.com; 4Oral and Maxillofacial Surgery Department, Universitat Internacional de Catalunya (UIC), St Josep Trueta s/n, 08195 Barcelona, Spain; 5Oral and Maxillofacial Surgery and Pediatric Dentistry Department, Universitat Internacional de Catalunya (UIC), St Josep Trueta s/n, 08195 Barcelona, Spain; eferresamat@uic.es; 6Biochemistry and Molecular Biology Department, Universidad Pública de Navarra, 31006 Pamplona, Spain; 7Inves Biofarm, Avd. Conocimiento, 34, 18016 Granada, Spain; anajimenez@invesbiofarm.com; 8Oral and Maxillofacial Surgery Department, Hospital Clinic de Barcelona, 08036 Barcelona, Spain

**Keywords:** BBL, THERAVEX, THERAVEX Tissue Care Plus, dental implant, bone bioactive liquid, Galaxy Titansure Active, Ziacom, oral wound healing, pain index score, ISQ, CBCTS

## Abstract

Implant stability can be compromised by factors such as inadequate bone quality and infection, leading to potential implant failure. Ensuring implant stability and longevity is crucial for patient satisfaction and quality of life. In this multicenter, randomized, double-blind clinical trial, we assessed the impact of a bone bioactive liquid (BBL) on the Galaxy TS implant’s performance, stability, and osseointegration. We evaluated the impact stability, osseointegration, and pain levels using initial stability quotient (ISQ) measurements, CBCT scans, and pain assessment post-surgery. Surface analysis was performed using scanning electron microscopy (SEM) and atomic force microscopy (AFM). In vitro studies examined the BBL’s effects on dental pulp pluripotent stem cells’ (DPPSCs’) osteogenesis and inflammation modulation in human macrophages. All implants successfully osseointegrated, as demonstrated by the results of our clinical and histological studies. The BBL-treated implants showed significantly lower pain scores by day 7 (*p* < 0.00001) and improved stability by day 30 (ISQ > 62.00 ± 0.59, *p* < 8 × 10^−7^). By day 60, CBCT scans revealed an increased bone area ratio in BBL-treated implants. AFM images demonstrated the BBL’s softening and wettability effect on implant surfaces. Furthermore, the BBL promoted DPPSCs’ osteogenesis and modulated inflammatory markers in human primary macrophages. This study presents compelling clinical and biological evidence that BBL treatment improves Galaxy TS implant stability, reduces pain, and enhances bone formation, possibly through surface tension modulation and immunomodulatory effects. This advancement holds promise for enhancing patient outcomes and implant longevity.

## 1. Introduction

Dental implantology presents patients with durable and natural-feeling solutions as an alternative to traditional dentures or bridges [1,2]. Implants are made from biocompatible materials, surgically inserted into the jawbone to mimic natural root structures and integrate with bone tissue [3,4]. This process creates a stable foundation for dental restorations [5,6] while preventing bone loss and preserving the facial structure [3,7,8]. These implants improve function, aesthetics, and quality of life, allowing patients to eat, speak, and smile confidently [9,10]. However, notably, inflammation presents a challenge for successful implantation, and pro-inflammatory cytokines such as interleukin-1B (IL-1β) and tumor necrosis factor-alpha (TNF-α) have been linked to bone resorption and osteoclastogenesis [11,12,13], although their roles in peri-implant diseases are still under investigation [14,15].

Implant stability, crucial for success, is often assessed using the implant stability quotient (ISQ), measured through resonance frequency analysis (RFA) [16,17]. ISQ values guide the timing of implant loading, with scores above 60 indicating good stability and scores below 50 suggesting a potential failure risk [16,18,19]. Monitoring ISQ values identifies problems early and enhances therapy success rates [16,20]. Factors influencing ISQ values and long-term osseointegration include bone density, surgical techniques, insertion torque, and implant design [21,22]. The healing duration before implant loading is critical, as premature loading can lower ISQ values and increase the failure risk [23]. Patient-related factors such as age, smoking, and systemic health conditions also affect stability and ISQ values [24,25].

In this article, we introduce a novel bone bioactive liquid (BBL, THERAVEX Tissue Care Plus) composed of phosphate saline solution (PBS) with calcium chloride and magnesium chloride (patents EP353211 and US 16/344,322). The BBL enhances cellular attraction at the bone–implant interface, increases surface hydroxyl groups, and improves hydration, thereby improving surface hydrophilicity and promoting active ionic interaction with blood plasma and bone progenitor cells for better tissue formation [26,27]. In pre-clinical animal models, BBL application demonstrated enhanced implant surface and bone regeneration [26]. Subsequently, we conducted a multicenter human clinical study with 33 participants and 264 implants to evaluate the BBL’s efficacy on Galaxy Titansure implants, assessing postoperative pain, ISQ values, CBCTs, bone formation, and implant surface morphology. Additionally, we studied the BBL’s impact on human dental pulp pluripotent stem cells (DPPSCs) and inflammatory markers in macrophages, seeking to understand its clinical effects.

## 2. Materials and Methods

### 2.1. Experimental Design

This multicenter, prospective clinical trial assessed the impact of BBL technology on Galaxy TS clinical implants (Galaxy Titansure, Ziacom Medical, Madrid, Spain), adhering to STROBE guideline ethical standards as per the revised Helsinki Declaration for biomedical research involving human subjects. Ethical approval was obtained from the ethics committee at Complejo Hospitalario de Toledo and the institutional review board of Spain (CEIm HM Hospitales 21.03.1786-GHM; protocol ID: V01–2021; date: 16 April 2021; Clinical Trial Registry Platform: Clinical Trial Gov. Press). Patients received a comprehensive explanation of the study protocol, signed an informed consent form, and provided written authorization to be included in the study population. All patients underwent uniform drilling sequencing, receiving implants of consistent dimensions. In addition, standardized implant diameters and lengths were utilized across all patients to minimize potential variables affecting outcomes. The URL and unique identification number for the trial registration are as follows: https://www.centerwatch.com/clinical-trials/listings/NCT06371430/bone-bioactive-liquid-efficiency-in-improving-dental-implant-osteointegration-oral-soft-tissue-hellingand-oral-surgery (accessed on 29 August 2024). Unique identification number: NCT06371430, Biointelligent Technology Syst.

Sample size calculation considered the implant torque (IT) as the primary variable. Several independent variables were assessed, including the operated arch, operated area, pain scale on days 1, 4, and 7, new bone formation observed through CBCTs on days 1 and 60 post-implantation, and ISQ values on days 1, 7, and 30 post-surgeries. Additionally, in vitro characterization of implant surfaces and osseointegration parameters utilized a DPPSC differentiation protocol, while pro- and anti-inflammatory cytokine levels were studied using human primary macrophages cultured on Galaxy TS or Galaxy TSA discs.

### 2.2. Study Population Inclusion and Exclusion Criteria

A thorough preliminary assessment included reviewing patients’ medical and dental histories, conducting detailed clinical examinations, and evaluating oral hygiene. Inclusion criteria for participants were an age of 18 or older, sufficient residual bone volume for implant placement without bone augmentation, a minimum ridge height and width of ≥9 mm and ≥6 mm, respectively, and healed bone sites with at least 1 month of post-extraction healing. Exclusion criteria included alcoholism, smoking, illicit drug use, heart diseases, diabetes, previous bone regenerative procedures, bleeding disorders, compromised immune systems, history of radiation therapy, and treatment with steroids or bisphosphonates. These criteria aimed to ensure a homogeneous patient population and minimize confounding factors affecting the investigation outcome.

### 2.3. Clinical Procedures

Galaxy TS implants (non-treated surface) and Galaxy TSA implants (BBL-treated surface) were categorized using a split-mouth study model. Implants were placed to replace the same tooth or teeth bilaterally, with one side receiving BBL treatment and the other side not. All implants were of the same dimension and osteotomy diameters for each patient, obtained from Ziacom Medical (Madrid, Spain). Patients were unaware of the area treated with BBL. Additionally, the dentists taking implant torque (IT) and implant stability quotient (ISQ) readings were blinded to the drilling protocol. At the study’s conclusion, all patients were informed of the results for both dental implant surfaces.

### 2.4. Surgical Technique

Before surgery, patients rinsed with 0.2% chlorhexidine for a minute to minimize the contamination risk from external sources. Sterile drapes covered the patient’s chest. Local anesthesia, either mepivacaine or articaine with epinephrine (1:100,000), was administered. Full-thickness surgical flaps were raised once anesthesia took effect. Implant osteotomies were performed with saline irrigation at 800 rpm, following the manufacturer’s recommendations for the Galaxy-Ziacom implant system. Implants were inserted at 20–50 rpm without irrigation using a manual torque wrench. Insertion torque (IT) was recorded. A total of 264 implants were placed in 33 patients, 160 post-extraction and 104 after one month. To ensure consistency and reliability in the outcomes, implants of the same length and platform were used uniformly in each patient. Specifically, within each patient, implants of identical length and platform were employed on both sides of the maxilla. This approach was consistently applied across all patients to maintain uniform conditions throughout this study. Sutures were typically removed during the postoperative follow-up appointment one week later.

Implant insertion began with a motor handpiece at 20–50 rpm without irrigation. A manual surgical torque wrench completed the process, recording the maximum torque value (Ncm) as the insertion torque (IT). A total of 264 implants were placed in 33 patients, including 160 post-extraction and 104 after at least one month. Sutures were usually removed during the postoperative follow-up appointment one week later.

Dental implants were placed in the central incisor region with a buccal orientation. This orientation is often necessary due to increased bone thickness on the buccal side, which facilitates implant insertion, and also offers stronger bone properties, such as a higher nanoindentation elastic modulus and plastic hardness compared to lingual bone tissues [28]. These properties contribute to enhanced implant stability. Additionally, the buccal orientation helps achieve better aesthetic outcomes, especially in the anterior region, where the implant’s position significantly influences the appearance of the gumline and the natural look of the restoration. This placement also ensures that the implant crown aligns more naturally with adjacent teeth, creating a more harmonious smile.

### 2.5. RFA and ISQ Measurements

After fully seating the implants, a Smartpeg specific to the implant system and restorative platform diameter was used for each implant. Resonance frequency analysis (RFA) was then conducted using an OsstellMentor device from Ostell/Integration Diagnostics in Gothenburg, Sweden. ISQ values were measured for all implant surfaces on days 0, 7, and 30 post-surgeries, with four readings taken per lingual, mesial, distal, and vestibular direction for each implant. The average ISQ values of all readings were recorded. Subsequently, new sterile healing abutments were inserted, and incisions were sutured to close the wounds.

### 2.6. CBCT Image Analysis Using ImageJ and Ilastik Programs

CBCT scans were taken for 24 implants in six patients, with 12 Galaxy TS (non-treated) implants on one side of the jaw and 12 Galaxy TSA (treated with BBL) implants on the other side. All patients received implants in the upper jaw using the “All on 4 Technique”, with 4 implants per patient. Scans were conducted on days 1 and 60 post-surgery using a 3D CS8100 CARESTREAM scanner (KODAK, Rochester, NY, USA). To ensure the consistency of the imaging parameters across all scans, we meticulously calibrated the scanner with the following technical settings: 90 kV, 4 mA, 87.5 mAs, and a voxel size of 150 µm, with a 360° rotation and a scanning time of 15 s, as previously described [29].

To maintain uniformity in the imaging process, a standardized bite mark was employed for midline positioning during each scan. This technique involved placing the patient in a fixed, repeatable position with the bite mark ensuring that the patient’s jaw alignment remained consistent across all imaging sessions. By using the bite mark to align the patient’s midline, we could ensure that each scan captured the same anatomical slice, thereby minimizing variability between scans. This method was crucial for achieving reliable longitudinal comparisons between the initial and follow-up scans.

CBCT images were analyzed as follows: Initially, images were cropped in ImageJ to specify the region of interest, with areas 20 pixels beyond the implant impact omitted. Images were then processed in Ilastik software (version 1.0) using machine learning algorithms [30]. The program was trained to segment four categories of preference areas. Batch processing was performed, and images were imported back to ImageJ for area measurement using the Glasby filter.

### 2.7. Procedures for Measurement of Post-Surgical Pain, Safety, and Discomfort

To evaluate the surgical procedure’s efficacy, pain scale assessments were conducted on days 1, 4, and 7 post-surgeries, involving direct patient contact to gauge subjective pain levels. A modified visual analog scale (VAS) was used, where no pain equated to 0, moderate pain to 5, and maximum pain to 10 [27,31]. Safety measurements were also considered, with us monitoring adverse events (AEs) and serious adverse events (SAEs) throughout this study. These events were either detected by the investigator or reported by patients, aiming to identify any potential complications arising from the procedure. By assessing efficacy through pain scale evaluations and safety through adverse event monitoring, this study aimed to evaluate outcomes and associated risks comprehensively.

### 2.8. Preparation of the Titanium Discs and DPPSCs’ Osteogenic Differentiation

Ziacom Medical provided titanium discs in sterile packaging, derived from commercially available titanium Galaxy implants and cut into 2.0 mm thick discs with a diameter of 14.0 mm. These discs were then divided into two experimental groups: the Galaxy TS, featuring a non-treated surface; and the Galaxy TSA, which were soaked in the BBL for 24 h before application.

Osteogenic differentiation of dental pulp pluripotent-like stem cells (DPPSCs) was conducted as previously described, with slight modifications [32,33,34]. DPPSCs at passage 5 (1 × 10^4^ cells/cm^2^) were seeded onto titanium discs sourced from Galaxy TS or Galaxy TSA and placed in 24-well plates. The osteogenic induction medium comprised α-Minimum Essential Eagle’s Medium (α-MEM, Invitrogen, Waltham, MA, USA), supplemented with 10% heat-inactivated fetal bovine serum (FBS, Invitrogen), 10 mM β-glycerophosphate (Sigma-Aldrich, St. Louis, MO, USA), 50 μM L-ascorbic acid (Sigma-Aldrich), 0.01 μM dexamethasone (Sigma-Aldrich), and 1% penicillin/streptomycin solution (Invitrogen). The medium was refreshed every 3–4 days for a total duration of 21 days.

### 2.9. Atomic Force Microscopy (AFM)

Samples were mounted onto flat Teflon discs using two-component epoxy (Nural 27, Pattex, Dusseldorf, Germany), with a 30 min curing period. Next, 200 µL of buffer solution was applied to sustain a liquid environment. The samples were then placed on the AFM sample plate, and AFM probes (SNL10 probes, nominal spring constant of 0.35 nN/nm) composed of silicon nitride were introduced into the liquid for 20 min to reach thermal equilibrium. This combination of materials provides both a low spring constant and sharpness (ca. 5 nm). Topographic imaging was conducted using the tapping mode at the minimum vertical force, with images captured at 512 × 512 pixels and 1 line/second. Simultaneously, phase imaging contrast was optimized by adjusting the amplitude setpoint of the AFM probe. This process was executed by an external service.

### 2.10. Scanning Electron Microscopy

Samples were immersed in 2.5% glutaraldehyde (Ted Pella Inc., Redding, CA, USA) in 100 mM na-cacodylate buffer (pH 7.2, EMS, Electron Microscopy Sciences, Hatfield, PA, USA) for 1 h on ice for fixation. Subsequently, they were treated with 1% osmium tetroxide (OsO_4_) for 1 h. Dehydration was carried out using serial solutions of acetone (30–100%), with the scaffolds mounted on aluminum stubs. Finally, the samples were examined using a Zeiss 940 DSM scanning electron microscope (Jena, Germany). This process was conducted by an external service.

### 2.11. Extracellular Calcium Accumulation and Quantification

The calcium concentration in the supernatant of differentiated DPPSCs was measured using a Calcium Colorimetric Assay Kit (BioVision, Zurich, Switzerland), following the manufacturer’s instructions. Standard solutions were utilized to calculate the calcium concentration, and absorbance was measured at 575 nm. Measurements were taken on days 7, 14, and 21 of differentiation, as described previously [32,35].

### 2.12. Alkaline Phosphatase (ALP) Activity

During DPPSC osteogenic differentiation on the discs, ALP activity was quantified on days 7, 14, and 21 using an Alkaline Phosphatase Kit (Applied Biosystems, Waltham, MA, USA), following the manufacturer’s instructions. Cells were fixed with 10% formalin-neutral buffer solution for 20 min. After three washes with 1 mL of MilliQ water, a chromogenic substrate (200 μL) was added, and incubation continued for 20 min. Optical density (OD) values were measured at 405 nm using a microplate reader at various time points—1, 2, 3, 4, and 5 min—as previously described [32,35].

### 2.13. Macrophage Cultures and Levels of Secreted Inflammatory Markers

Human primary macrophages (STEMCELL Technologies, Vancouver, BC, Canada) were cultured in RPMI 1640 (ThermoFisher Scientific, Waltham, MA, USA) supplemented with fetal bovine serum, penicillin–streptomycin, and macrophage colony-stimulating factor (M-CSF, BioLegend, San Diego, CA, USA) to generate macrophages. After seven days, macrophages (200,000 cells/cm^2^) were passaged and plated onto Galaxy TS or Galaxy TSA discs; the latter were soaked in the BBL for 24 h before cell culture, as previously described [35]. Following attachment, discs were placed in culture vials containing media and incubated for 48 h. Secreted pro-inflammatory (IL-1B and TNF-a) and anti-inflammatory (IL-4 and IL-10) cytokines were quantified in conditioned media using a custom human multiplex ELISA panel (LEGEND-plex, BioLegend), with protein levels normalized to the total DNA content using a QuantiFluor dsDNA System (Promega, Madison, WI, USA).

### 2.14. Statistical and Analytical Methods

The normality of the distribution was assessed using the Shapiro–Wilk normality test, as previously described [27]. Data were presented as the mean and standard error of the mean. A two-way ANOVA with a significance level of *p* ≤ 0.05 was employed [36]. Data analysis was performed using the Statistical Package for Social Sciences software (SPSS, version 23, IBM Corp., Armonk, NY, USA).

## 3. Results

### 3.1. BBL Treatments Reduced VAS

Following the surgical procedures, both the BBL-treated and control groups exhibited moderate pain responses based on the VAS index on days 1, 4, and 7 post-surgery (Table 1). Statistical analysis revealed a significant improvement in VAS scores at all time points for the BBL treatment groups compared to their respective controls (*p* = 1.6 × 10^−7^ and *p* = 1 × 10^−5^ for the upper and lower jaws, respectively; Table 1), indicating a positive impact of BBL treatments on pain reduction.

### 3.2. BBL Treatments Improved the ISQ Values

A comparison of ISQ values between BBL-treated and control Galaxy implants showed comparable values on day 1 post-surgery, but significant differences emerged by day 7 (Table 2). BBL-treated implants demonstrated notably higher ISQ values at both upper jaw (*p* = 0.022) and lower jaw (*p* = 0.002) positions (Table 2). By day 30, BBL-treated implants exhibited a remarkable increase in ISQ values compared to controls, with statistically significant improvements at both positions (*p* ≥ 5 × 10^−7^). A visual representation of the individual ISQ values corroborated these findings (Figure 1).

### 3.3. CBCT Data Analysis

CBCT scan images comparing 1 day to 60 days post-surgeries revealed a significant increase in newly generated bone on BBL-treated Galaxy implants relative to untreated surfaces, indicating enhanced osseointegration (Figure 2A,B). This methodology allowed for the precise assessment of new bone formation on the implant surfaces, providing valuable insights into the effectiveness of BBL treatment in promoting osseointegration.

### 3.4. Titanium Discs’ Characterization

The surface characterization of MIS TS and MIS TSA discs was performed using SEM and AFM. SEM images of the MIS TS disc surface at various magnifications revealed a consistent “uniform-rough” texture (Figure 3A). The SEM analysis revealed that MIS surfaces exhibit a distinct topographical pattern characterized by a combination of macro-textured roughness due to sandblasting with large-grit particles and micro-textured features resulting from acid etching. Sandblasting imparts a rough, uneven texture with noticeable grooves and ridges, while the subsequent acid etching process creates a porous, micro-rough surface.

AFM measurements were employed to evaluate the surface morphology of MIS TS and MIS TSA. The 3D topography images of MIS TS showed considerably higher roughness, measuring approximately 100 nm (Figure 3(Ba)), in comparison with the BBL-treated surface disc, which exhibited a roughness of about 33 nm (Figure 3(Ca)). These differences in the surface characteristics are further illustrated in the 2D images, where variations in the color depth levels are evident (Figure 3(Bb,Cb)). Notably, the BBL treatment creates a liquid environment that reduces the disc roughness and stiffness, as visualized in the AFM mechanical mapping (Figure 3(Cc)), in comparison to untreated MIS TS discs (Figure 3(Bc)).

### 3.5. DPPSC Osteogenic Differentiation on Discs

To assess the impact of BBL treatment on implant-disc surface modifications and subsequent bone regeneration, DPPSCs were directed towards osteogenic lineage on BBL-treated titanium discs compared to untreated ones. The efficiency of differentiation and functional activity were evaluated via SEM, calcium mineralization, and ALP activity.

SEM images revealed that DPPSCs on BBL-treated discs exhibited bone-like architecture, with compact structures observed (Figure 4B,D). In contrast, non-treated discs showed less pronounced bone-like structures (Figure 4A,C). Moreover, calcium nodules were prominently formed on BBL-treated discs, indicating enhanced mineralization (Figure 4D). Importantly, osteogenic differentiation on BBL-treated discs surpassed that on untreated discs.

The functional activity of differentiated DPPSCs was assessed through extracellular calcium secretion and ALP activity assays. DPPSCs on BBL-treated discs exhibited higher calcium secretion over time compared to untreated discs, indicating enhanced mineralization (Figure 4E). Similarly, ALP activity was significantly higher in DPPSCs cultured on BBL-treated discs compared to untreated discs (Figure 4F).

### 3.6. Macrophages’ Inflammatory Marker Levels

These findings collectively highlight the superior efficacy of BBL-treated surfaces in promoting osteogenic differentiation and reducing pain, suggesting a potential mechanism for the enhanced bone regeneration observed in patients receiving BBL-treated implants. Building upon this, we hypothesized that BBL treatment may modulate inflammatory factors or cell survival. To test this hypothesis, human primary macrophages were cultured on both Galaxy TS (non-treated surface) and Galaxy TSA (BBL-treated surface), and inflammatory factor levels were quantified using ELISAs.

Figure 5 illustrates a significant reduction in the secretion of pro-inflammatory cytokines, IL-1β and TNF-α, in primary macrophages cultured on Galaxy TSA discs compared to non-treated Galaxy discs. Conversely, macrophages cultured on non-treated discs exhibited lower levels of the anti-inflammatory markers IL-4 and IL-10. These findings suggest that BBL treatment exerts a regulatory effect on inflammatory factors, potentially influencing the cellular environment and contributing to the observed enhancement in osteogenic differentiation, pain reduction, and subsequent bone regeneration in BBL-treated implants.

## 4. Discussion

The findings of this study shed light on the interaction between the BBL and Galaxy TS surfaces and MIS TS, contributing to ongoing efforts to enhance the bioreactivity of SLA (sandblasted, large-grit, acid-etched) surfaces, known for their favorable roughness characteristics for implantation [37]. While SLA surfaces have demonstrated good osseointegration properties, further improvement through surface modifications using bioactive materials has been explored extensively [38,39,40].

In our previous animal study, we observed enhanced bone apposition during the early stages of bone formation when using SLA titanium surfaces treated with BBL compared to untreated control implants [26]. In this study, we demonstrate a significant increase in mineralized bone-to-implant contact following a short healing period with BBL-treated SLA surfaces, the opposite to untreated controls. Notably, BBL treatment improves the surface wettability and roughness of the treated Galaxy TS implants, which can most likely be attributed to the improvement of early osseointegration. It is well documented that the liquid bioactive material improves implant wettability and reduces roughness; however, the reactivity of liquid bioactive surfaces depends on several factors, including their hydrophilic properties, salt composition and concentration, electrodynamic charge, absence of solid contaminants, and absence of proteins [41,42]. These factors can influence the viscosity, stiffness, and roughness of the implants, thereby affecting their performance in clinical applications.

Moreover, primary human macrophages cultured on BBL-treated Galaxy TSA discs exhibited lower pro-inflammatory and higher anti-inflammatory cytokine secretion levels than cells on non-treated Galaxy TS discs. Maintaining a balance between pro-inflammatory (IL-1β and TNF-α) and anti-inflammatory (IL-4 and IL-10) mediators is essential for optimal osseointegration [43,44]. An imbalance in these factors could potentially lead to destructive reactions, exacerbating peri-implant disease progression [45]. Investigating these dynamics further could provide insights into peri-implant diseases and aid in developing targeted therapeutic strategies.

The BBL enhances bone differentiation by optimizing the surface characteristics of implants, particularly surface roughness, and wettability, which are critical factors for cellular adhesion and osteogenesis. Increased surface roughness provides a larger contact area for osteoprogenitor cells, such as osteoblasts, to adhere, promoting their proliferation and differentiation into bone-forming cells. Enhanced wettability improves the implant’s interaction with biological fluids, further supporting cell attachment and spreading. In parallel, BBL modulates the initial immune response by orchestrating a shift in cytokine profiles, promoting anti-inflammatory cytokines such as IL-4 and IL-10. This shift fosters a controlled inflammatory environment conducive to tissue repair, minimizing excessive inflammation and the peri-implantitis risk. Achieving this precise balance between inflammation and regeneration is fundamental for successful osseointegration and long-term implant stability. Furthermore, we do not exclude the possibility that the BBL could also possess antibacterial properties through the activation of reactive oxygen species (ROS), as previously described for the action of copper sulfide molecules on graphitic carbon nitride (CuS@g-C_3_N_4_) nanoparticles [46]. The antibacterial properties of BBL will be further investigated.

ISQ values are crucial in evaluating implant primary stability, with higher values signifying the heightened stability essential for successful osseointegration. ISQ values also indicate reduced micromotion, which is critical for immediate loading protocols [47,48,49,50]. Additionally, they enable clinicians to monitor stability throughout the healing process [16], aiding in treatment decisions and optimizing conditions for enhanced osseointegration [51], thus improving clinical outcomes.

In this study, the split-mouth design with 33 participants and 264 implants mitigated potential confounding factors [52]. While both Galaxy TS and Galaxy TSA implants demonstrated acceptable primary stability, notable differences emerged in secondary stability, emphasizing the potential impact of surface treatment on implant performance during the initial healing phase. Long-term follow-up is necessary if we are to fully understand the implications. Our study aimed to assess CBCT’s diagnostic accuracy in detecting new bone formation on dental implant surfaces 60 days post-insertion. CBCT showed a higher sensitivity than digital periapical radiography, facilitating a precise assessment of bone formation dynamics and contributing to a comprehensive understanding of implant integration processes [53,54,55,56].

## 5. Study Limitations

The evaluation of osseointegration often relies on surrogate variables such as CBCT and/or ISQ measurements, which offer useful but limited information. CBCT offers a non-invasive means to assess bone density and proximity of the implant to surrounding structures, but it lacks the ability to directly assess the quality of the bone-to-implant interface at a microscopic level [57,58]. Similarly, ISQ measures the mechanical stability of an implant, but does not provide information on the biological integration processes occurring at the cellular level [57]. These surrogate measures, although practical for clinical use, may fail to capture the true extent of osseointegration, which can only be definitively determined through histological analysis that allows for direct observation of bone formation and remodeling around the implant. Therefore, reliance on these indirect methods may lead to over- or underestimation of the actual osseointegration, impacting the long-term success and prognosis of dental implants.

## 6. Conclusions

In conclusion, our study highlights the potential of BBL (THERAVEX Tissue Care Plus) -treated surfaces to enhance osseointegration and improve clinical outcomes in implant dentistry. Surface modifications, such as BBL treatment, hold promise in optimizing the bioreactivity of SLA surfaces, leading to increased mineralized bone/implant contact and improved stability during the initial healing phase. Additionally, BBL treatment appears to modulate inflammatory factors, promoting a balanced immune response conducive to optimal osseointegration. While our study provides valuable insights, further research with larger sample sizes and long-term follow-up is needed to validate these findings and refine clinical practices in implant dentistry.

## Figures and Tables

**Figure 1 jfb-15-00293-f001:**
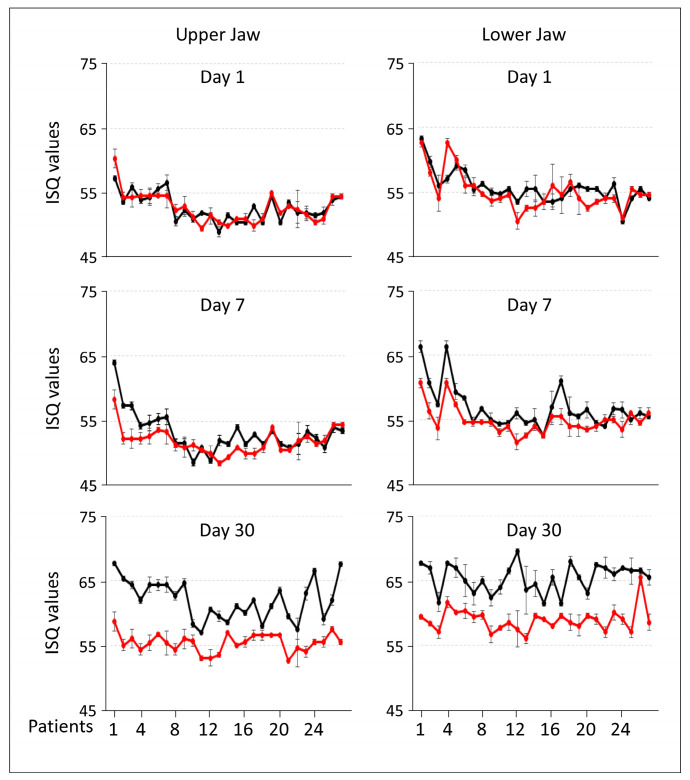
The mean implant stability quotient (ISQ) values for the upper and lower jaws for each patient individually measured on days 1, 7, and 30 post-surgeries. The implant stability was measured via resonance frequency analysis. Each dot represents the ISQ value/patient. Galaxy TSA implants treated with BBL are in black, and Galaxy TS implants without BBL are in red. The values are means ± standard errors of the means for four readings.

**Figure 2 jfb-15-00293-f002:**
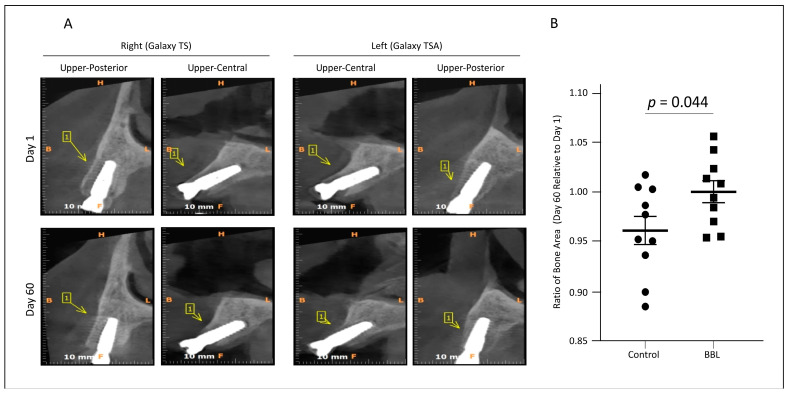
CBCT image data analysis. (**A**) Representative CBCT images from an individual taken on day 1 and day 60 post-surgery. Implants at the right upper posterior and central sites were treated with control Galaxy TS, while implants at the left upper posterior and central sites were treated with BBL Galaxy TSA. (**B**) The ratio of generated bone on day 60 relative to that on day 1 was calculated as described in the Materials and Methods section. The values are means ± standard errors of the means for four readings.

**Figure 3 jfb-15-00293-f003:**
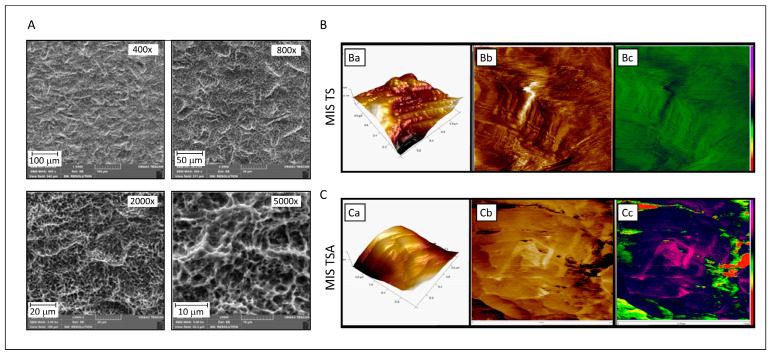
Characterization of titanium alloy discs of the MIS implant surface. (**A**) SEM analysis of MIS TS implant at different magnifications (400×, 800×, 2000×, and 5000×) after alumina blasting and acid etching. (**B**,**C**) AFM analysis of discs of MIS TS and MIS TSA surfaces, respectively. (**Ba**) MIS TS surface: 1 × 1 micron 3D topographic image (surface roughness Rq parameter is 1.3 nm) showing homogeneity on the sample surface. (**Bb**,**Bc**) MIS TS surface: 1 × 1 micron 3D topographic image (surface roughness is 1.3 nm), where no substrate structure can be observed on the surface. (**Ca**) MIS TSA surface: 1 × 1 micron 3D topographic image (surface roughness Rq parameter is 1.3 nm) and 1 × 1 micron phase images (in two different color scales) corresponding to the previous topographic image, showing homogeneity on the sample surface. (**Cb**,**Cc**) MIS TS surface: 1 × 1 micron 3D topographic image (surface roughness is 74.5 nm) and 1 × 1 micron phase images corresponding to the previous topographic image. The surface appears less rough than in the blank sample, implying that the substrate structure is coated with BBL, which reduces the smoothness.

**Figure 4 jfb-15-00293-f004:**
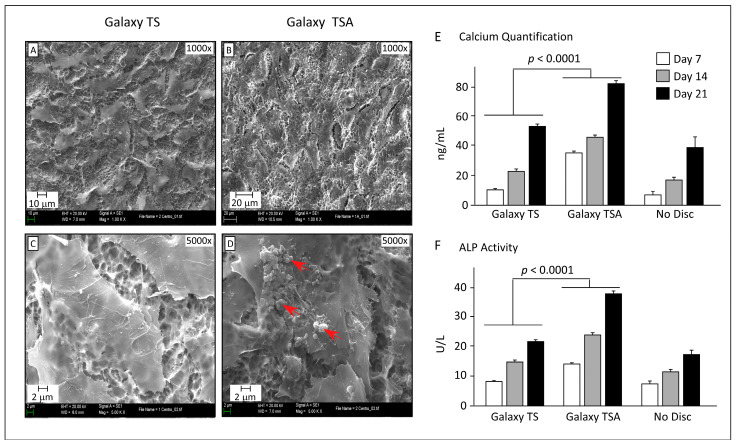
Evaluation of the osteogenic differentiated DPPSCs cultured on Galaxy TS (non-treated surface) and Galaxy TSA (BBL-treated surface) discs. (**A**–**F**): Representative SEM images for the osteogenic differentiated DPPSCs: Galaxy TS (non-treated surface, **A**–**C**) and Galaxy TSA (BBL-treated surface, **D**–**F**) discs. Images were taken at different magnifications: 1000× and 5000×. (**E**): Extracellular calcium quantification (µg/µL) on days 7, 14, and 21 of DPPSC osteogenic differentiation, red arrows indicate calcium nodules formation and mineralization. (**F**): ALP activity (U/L) on days 7, 14, and 21 of osteogenic differentiation. Together, these data indicate that BBL-treated surfaces enhance osteogenic differentiation.

**Figure 5 jfb-15-00293-f005:**
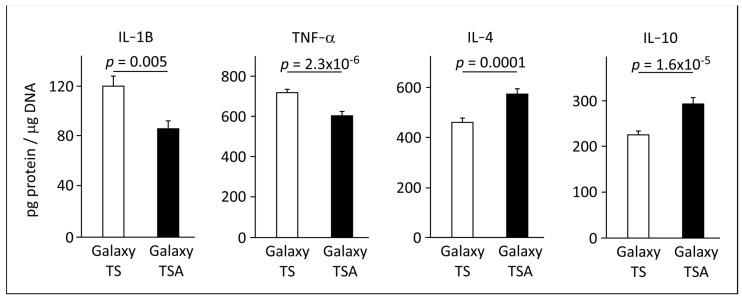
Secretion levels of inflammatory cytokines from human primary macrophages cultured on Galaxy TS and Galaxy TSA discs. BBL treatments exhibited significant reductions in the levels of secreted pro-inflammatory markers IL−1B and TNF−α, with a concurrent elevation in the levels of secreted anti-inflammatory markers IL−4 and IL−10 from primary human macrophages. The observed modulation in cytokine secretion suggests that BBL treatment may play a role in creating an anti-inflammatory microenvironment.

**Table 1 jfb-15-00293-t001:** Pain index VAS, average, and SEM for the 27 patients for each treatment.

Index	Upper Jaw			Lower Jaw		
	Galaxy TSA	Galaxy TS	*p*-Value	Galaxy TSA	Galaxy TS	*p*-Value
Pain	1.93 ± 0.39	3.41 ± 0.28	1.60 × 10^−7^	1.96 ± 0.30	2.85 ± 0.31	0.00001

**Table 2 jfb-15-00293-t002:** Implant stability quotient (ISQ) mean values on days 1, 7, and 30 post-surgeries.

Time	Upper Jaw			Lower Jaw		
	Galaxy TSA	Galaxy TS	*p*-Value	Galaxy TSA	Galaxy TS	*p*-Value
Day 1	52.8 ± 0.41	52.72 ± 0.45	0.943	55.65 ± 0.45	55.02 ± 0.55	0.397
Day 7	53.18 ± 0.58	51.90 ± 0.38	0.020	56.88 ± 0.63	54.86 ± 0.40	0.010
Day 30	62.00 ± 0.59	55.38 ± 0.28	5 × 10^−8^	65.38 ± 0.41	58.89 ± 0.38	8 × 10^−7^

## Data Availability

The original contributions presented in the study are included in the article, further inquiries can be directed to the corresponding author.

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
