# Peer review of "Randomized Clinical Trial: Bone Bioactive Liquid Improves Implant Stability and Osseointegration"

_jfb, 2024, doi:10.3390/jfb15100293_

Round 1
Reviewer 1 Report
Comments and Suggestions for Authors
The authors of the paper discuss a very interesting research study. The bone bioactive liquid impact on Galaxy TS implants performance, stability and osseointegration was adequately studied, assessed and well described. The evidences are clear and the treatment improves implant stability, reduces pain and enhances bone formation.
The Introduction, although short, identifies the main published papers concerning the objective of the study. The Materials and Methods section is adequate, very well described and is the strongest part of the paper. All points of this section are clear concerning the aim of each one.
In the Results section I suggest the separation of the images from the graphics. The graphic of figure 2 is relatively small. The Discuss section, although short, includes very selective and appropriate references. Conclusions are adequate.
In the title of the paper authors use osteointegration and within the text use osseointegration.
A very interesting research paper, well-structured and research conducted. I strongly advise publication.
Author Response
Comments and Suggestions for Authors
The authors of the paper discuss a very interesting research study. The bone bioactive liquid impact on Galaxy TS implants performance, stability and osseointegration was adequately studied, assessed and well described. The evidences are clear and the treatment improves implant stability, reduces pain and enhances bone formation.
The Introduction, although short, identifies the main published papers concerning the objective of the study. The Materials and Methods section is adequate, very well described and is the strongest part of the paper. All points of this section are clear concerning the aim of each one.
In the Results section I suggest the separation of the images from the graphics. The graphic of figure 2 is relatively small. The Discuss section, although short, includes very selective and appropriate references. Conclusions are adequate.
In the title of the paper authors use osteointegration and within the text use osseointegration.
A very interesting research paper, well-structured and research conducted. I strongly advise publication.
We would like to thank the reviewer for his/her feedback and complaints. We fixed figures 2 and 4 to better visualize the graphs, but we find it difficult to separate as the data are related to the images part of the figure. We agree that the corrected terminology is osseointegration, therefore we corrected the title and all related terminologies.

Reviewer 2 Report
Comments and Suggestions for Authors
Comments:
In this multi-center 34 randomized, double-blind clinical trial, a bone bioactive liquid (BBL) im-35 pact on Galaxy TS implants performance, stability and osseointegration. This study presents compelling clinical and biological evidence that BBL treat-48 ment improves Galaxy TS implant stability, reduces pain, and enhances bone formation, 49 possibly through surface tension modulation and immunomodulatory effects. This research work is very interesting, and the reviewer has some questions and suggestions that the author should take on board.
1. Experimental supplementation of osteogenic differentiation
2. The author should supplement the mechanism of BBL's on osteogenic and inflammatory regulation of human macrophage dental pulp pluripotent stem cells (DPPSCs).
3. The paper may be helpful for enriching the content of the manuscript. (doi.org/10.1016/j.colsurfb.2023.113512)
4. English is very poor and recommended to seek help from experts to modify English.
Comments on the Quality of English LanguageEnglish is very poor and recommended to seek help from experts to modify English.
Author Response
Reviewer 2
Comments and Suggestions for Authors
In this multi-center randomized, double-blind clinical trial, a bone bioactive liquid (BBL) impact on Galaxy TS implants performance, stability and osseointegration. This study presents compelling clinical and biological evidence that BBL treatment improves Galaxy TS implant stability, reduces pain, and enhances bone formation, possibly through surface tension modulation and immunomodulatory effects. This research work is very interesting, and the reviewer has some questions and suggestions that the author should take on board.
We would like to thank the reviewer for his/her feedback and complaints.
- Experimental supplementation of osteogenic differentiation
We would like to thank the reviewer for the valuable feedback. We have revised and improved the explanation of the osteogenic differentiation protocol for DPPSC cells in the M&M manuscript (Please see Pages 5 and 6, Lines 207-215), providing more detailed information about the experimental steps and conditions. We would like to emphasize that this protocol is a standard method widely used in both our previous work and the research of other scientists, as reflected in the cited bibliographic references. We have updated the relevant citations to provide further context and support for our methodology.
- The author should supplement the mechanism of BBL's on osteogenic and inflammatory regulation of human macrophage dental pulp pluripotent stem cells (DPPSCs).
We would like to thank the reviewer for valuable feedback. We have improved the explanation of the mechanism of action of BBL in bone differentiation and the inflammatory process as requested. This explanation has been incorporated into the Discussion section (Please see Page 14, Lines 432-443) of the manuscript to provide clearer insights into how BBL modulates implant surface properties and the immune response, which are crucial for effective osseointegration. We believe these additions enhance the overall clarity and depth of our findings.
- The paper may be helpful for enriching the content of the manuscript. (doi.org/10.1016/j.colsurfb.2023.113512)
We would like to thank the reviewer for valuable feedback. We find the publication interesting. Although we have not tested the role of BBL on ROS activation and antibacterial properties, but we cannot exclude such a role (Please see page 14, lines 433-448).
- English is very poor and recommended to seek help from experts to modify English.
The manuscript will be sent for English Language Editing after the final version approval.
Reviewer 3 Report
Comments and Suggestions for Authors
This is a highly novel and interesting paper.
However, some corrections are needed to clarify the content with regard to methods and results. Please consider.
Methods.
1. there are no descriptions or photographs of the shape and surface properties of the Galaxy implants used in this paper. Please correct this as it is an important experimental implant in this paper.
2. lacks information on the indicated patients and defects and the length of the implants.
 The number of cases is small, so please describe them clearly.
3. the BBL treatment method itself is not described in the first place. As this is the main objective of this study, please describe it in detail with references.
4. please describe in detail the measurement method of CBCT with images and references

Results.
1. maxilar suprior ? Please correct the meaning of the word ‘inferior’ in English, as I don't understand it.
Comments on the Quality of English Languagemaxilar suprior ? Please correct the meaning of the word ‘inferior’ in English, as I don't understand it.
Author Response
Comments and Suggestions for Authors
This is a highly novel and interesting paper.
However, some corrections are needed to clarify the content with regard to methods and results. Please consider.
We would like to thank the reviewer for the valuable complement and feedback.
Methods.
- There are no descriptions or photographs of the shape and surface properties of the Galaxy implants used in this paper. Please correct this as it is an important experimental implant in this paper.
We would like to thank the reviewer for valuable feedback. We have revised and improved the explanation of shape and surface properties of the Galaxy implants; this explanation has been incorporated into the result section (Please see Page 10, lines 325-330).
- lacks information on the indicated patients and defects and the length of the implants.
 The number of cases is small, so please describe them clearly.
We would like to thank the reviewer for valuable feedback. We have revised and improved the explanation of the indicated patients and defects and the length of the implants into M&M section (Please see Page 4, lines 144-148).
In our study, a total of 33 patients underwent surgical procedures, resulting in the placement of 264 dental implants. To ensure consistency and reliability in the outcomes, implants of the same length and platform were used uniformly in each patient. Specifically, within each patient, implants of identical length and platform were employed on both sides of the maxilla. This approach was consistently applied across all patients to maintain uniform conditions throughout the study (Please see Page 5, lines 169-181).
- the BBL treatment method itself is not described in the first place. As this is the main objective of this study, please describe it in detail with references.
We would like to thank the reviewer for the valuable feedback. The Galaxy TSA implants, provided by Ziacom Medical, are pre-soaked in BBL and ready for use. The discs used for stem cell differentiation and macrophage cultures were soaked in BBL for 24 hours prior to the commencement of the experimental procedures Please see Page 6, lines 205-206, and Page 7, Line 260).
- please describe in detail the measurement method of CBCT with images and references
We would like to thank the reviewer for the valuable feedback. We have revised and improved the explanation of the measurement method of CBCT in the M&M section (Please see Page 5, lines 169-181).

Results.
- maxilar suprior ? Please correct the meaning of the word ‘inferior’ in English, as I don't understand it.
We would like to thank the reviewer for the valuable feedback. We meant by maxilar superior the upper jaw and by maxilar inferior. To avoid any confusion for the reader, we corrected the terminology to upper and lower jaws at the result section (Please see Lines 282 and 292, Table 1 and 2, and Figure 1).
Please be advised that the manuscript will be submitted for English Language Editing after the review process.
Round 2
Reviewer 2 Report
Comments and Suggestions for Authors
The author has provided sufficient explanations for the question, which can be further published
Author Response
We would like to thank the Reviewer for the valuable insights and suggestions that improved our study.
Reviewer 3 Report
Comments and Suggestions for Authors
It has been appropriately modified.
The content is acceptable.
Author Response

(The authors gave the same response as above.)
